# Introducing Common Null Space of Gradients for Gradient Projection Methods in Continual Learning

Chengyi Yang
Shanghai Institute of AI for
Education, School of Computer
Science and Technology
East China Normal University
Shanghai, China
52265901027@stu.ecnu.edu.cn

Mingda Dong
Shanghai Institute of AI for
Education, School of Computer
Science and Technology
East China Normal University
Shanghai, China
52215901032@stu.ecnu.edu.cn

Xiaoyue Zhang
School of Statistics and Information
Science
Shanghai University of International
Business and Economics
Shanghai, China
22350014@suibe.edu.cn

Jiayin Qi
Cyberspace Institute of Advanced
Technology
Guangzhou University
Guangzhou, China
qijiayin@139.com

Aimin Zhou*
Shanghai Institute of AI for
Education, School of Computer
Science and Technology
East China Normal University
Shanghai, China
amzhou@cs.ecnu.edu.cn

## Abstract

Continual learning aims to learn new knowledge from a sequence of tasks without forgetting. Recent studies have found that projecting gradients onto the orthogonal direction of task-specific features is effective. However, these methods mainly focus on mitigating catastrophic forgetting by adopting old features to construct projection spaces, neglecting the potential to enhance plasticity and the valuable information contained in previous gradients. To enhance plasticity and effectively utilize the gradients from old tasks, we propose Gradient Projection in Common Null Space (GPCNS), which projects current gradients into the common null space of final gradients under all preceding tasks. Moreover, to integrate both feature and gradient information, we propose a collaborative framework that allows GPCNS to be utilized in conjunction with existing gradient projection methods as a plug-and-play extension that provides gradient information and better plasticity. Experimental evaluations conducted on three benchmarks demonstrate that GPCNS exhibits superior plasticity compared to conventional gradient projection methods. More importantly, GPCNS can effectively improve the backward transfer and average accuracy for existing gradient projection methods when applied as a plugin, which outperforms all the gradient projection methods without increasing learnable parameters and customized objective functions. The code is available at https://github.com/Hifipsysta/GPCNS.

*Aimin Zhou is the corresponding author.

## CCS Concepts

• **Computing methodologies → Lifelong machine learning**.

## Keywords

Continual Learning, Gradient Projection, Null Space, Gradient Information

**ACM Reference Format:**
Chengyi Yang, Mingda Dong, Xiaoyue Zhang, Jiayin Qi, and Aimin Zhou. 2024. Introducing Common Null Space of Gradients for Gradient Projection Methods in Continual Learning. In *Proceedings of the 32nd ACM International Conference on Multimedia (MM '24), October 28-November 1, 2024, Melbourne, VIC, Australia.* ACM, New York, NY, USA, 9 pages. https://doi.org/10.1145/3664647.3680605

## 1 Introduction

Machine learning algorithms lack the ability to continually learn new knowledge like humans. Specifically, a neural network will perform less well on old tasks after learning a new task, and this phenomenon is known as catastrophic forgetting (CF) [10, 24]. To address this issue, continual learning (CL) [30] has been proposed and applied to computer vision, natural language processing and multi-modal scenarios, including object detection [8, 22, 36, 44], semantic segmentation [9, 25, 35, 46], relation extraction [4, 13, 34], neural machine translation [21, 29, 43], cross-modal retrieval [38, 41] and visual question answering [16, 26, 42].

Continual learning considers the scenario of learning a stream of task-specific data sequentially, expecting to retain the knowledge related to old tasks when learning current task. Recently, gradient projection methods [3, 20, 27, 28, 33] have found that catastrophic forgetting can be effectively alleviated if the gradients under current task are orthogonal to the subspaces spanned by features from old tasks. These methods all project gradients onto the orthogonal direction of previous features, which have been demonstrated to be equivalent and hold a unified training paradigm called feature space continual learning paradigm (FSCLP) [45].

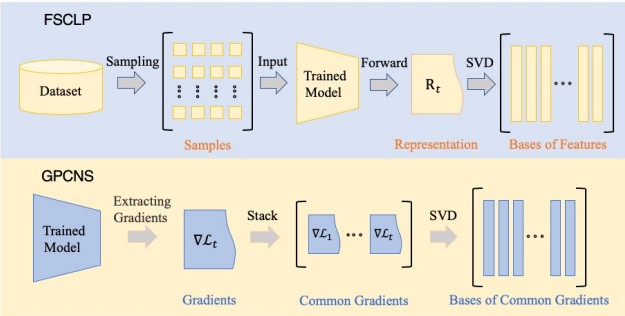

**Figure 1: FSCLP: Constructing a projection space through representation matrix, which is originated from collecting a sufficient number of (e.g. 100) samples from the dataset. GPCNS: Constructing a projection space throgh common gradient matrix, which is obtained after stacking final gradients from all previous task. We believe that feature and gradient information are equally important, and utilizing them simultaneously can construct better projection spaces.**

However, FSCLP methods primarily rely on features from previous tasks for constructing projection spaces, neglecting the valuable information contained in previous gradients. In addition, we note that the design idea of FSCLP approaches lies in alleviating catastrophic forgetting through minimizing parameter shifting, which is closely related to stability. Although these methods have proven effective in enhancing stability, they lack sufficient consideration in improving plasticity.

To improve plasticity and effectively utilize the gradients from old tasks, we attempt to demonstrate that gradients from previous tasks are equally powerful and important as previous features when applied to construct projection spaces. We cannot simply replace the features under FSCLP with gradients, because we have found that existing algorithms designed for features cannot fully mine gradient data (see Table 4). Therefore, we propose a novel approach called Gradient Projection in Common Null Space (GPCNS), which adopts a novel training paradigm as shown in the lower half of Figure 1. More importantly, since the design concept and information source of GPCNS are different from FSCLP, these two types of methods can be applied simultaneously to construct a better projection space. Motivated by this insight, we propose a dual information fusion framework to utilize both features and gradients from previous tasks. The contributions of this paper can be summarized as follows:

(1) We analyze that the effectiveness of existing methods under FSCLP lies in minimizing parameter shifting, which only focuses on stability. To improve plasticity, we suggests that gradients under different tasks should be projected in mutually orthogonal subspaces.

(2) We present a novel concept for projection space called common null space, which has rigorous mathematical proofs and geometric interpretation. In common null space, the mutual interference between gradients under different tasks can be minimized.

(3) We propose GPCNS which projects gradients into the common null space of final gradients under all previous tasks. GPCNS effectively utilizes the gradient information of old tasks and posses

higher plasticity (see Section 4.4), which are exactly what existing FSCLP methods lack.

(4) We propose a collaborative framework for FSCLP and GPCNS, which integrates feature and gradient information to construct a better projection space. GPCNS effectively compensates for the shortcomings of FSCLP methods in utilizing gradient information and plasticity.

## 2 Related Work

### 2.1 Gradient Projection Methods in CL

We focus on the gradient projection methods in continual learning, which project gradients onto the orthogonal direction of the feature space related to previous tasks. Orthogonal Weight Modification (OWM) [40] constructs projection operators through recursive least squares algorithm. However, OWM exhibits unstable performance due to its low backward transfer. Adam-NSCL [33] projects gradients into the null space of uncentered covariances of features, but the higher computational cost limits its application. Gradient Projection Memory (GPM) [27] stores the bases of features for each task in a memory buffer, and then projects the gradients onto the orthogonal direction of old features. Subsequent gradient projection methods can be seen as the improvements based on GPM. Class Gradient Projection (CGP) [3] computes projection subspaces from classes instead of from tasks. Trust Region Gradient Projection (TRGP) [20] and Scaled Gradient Projection (SGP) [28] enhance GPM by scaling the bases of feature spaces according to task similarity and importances respectively. In addition, Data Augmented Flatness-aware Gradient Projection (DFGP) [37] and Connector [18] redesign the objective function in GPM and Adam-NSCL respectively, which are essentially *regularization based methods*.

### 2.2 Gradient Information in CL

Gradient Episodic Memory (GEM) [23] and Averaged GEM (A-GEM) [1] adjust the current gradients based on the gradients computed with data in the memory. OGD [7] projects the new gradients orthogonally to the memorized gradients through Gram-Schmidt process, but it does not consider features and fail to achieve high performance. TRGP [20] and Adaptive Plasticity Improvement (API) [17] utilize the length of gradient projection to measure the correlation between tasks and determine whether to expand the network respectively. CUBER [19] and DFGP [37] utilize gradient information to design objective functions that encourage feedback knowledge transfer and flatter loss landscapes respectively. However, they all consider the gradients of old tasks as auxiliary information. Moreover, FS-DGPM [5] and CGP [3] have mentioned gradient subspaces, but they refer to the projection space for current gradient, rather than constructing the projection space through previous gradients. To our knowledge, this is the first work that gradient and feature information are considered equally important for constructing projection spaces in continual learning.

## 3 Methods

In this section, we analyze the causes of catastrophic forgetting and the effectiveness of FSCLP. Then we propose GPCNS and a collaborative framework for GPCNS and FSCLP respectively.

## 3.1 Training Paradigm of Existing Methods

**Continual Learning.** We consider the setting of supervised continual learning. A neural network $f_W$ with $W$ as its parameter tensor will sequentially learn a stream of data $\mathcal{D} = \{\mathcal{D}_1, \mathcal{D}_2, \cdots, \mathcal{D}_T\}$, where $t \in \{1, 2, \cdots, T\}$ is the task descriptor. $\mathcal{D}_t = \{(X_{t,i}, y_{t,i})\}_{i=1}^{N_t}$ represents the training data under task $t$, and $N_t$ is the data size. $X_{t,i}$ and $y_{t,i}$ are the $i$-th input data and label under task $t$, respectively. The method described below is for a certain layer in a network, which can be generalized to all the layers, thus we omit the layer notation $l$.

**Forgetting Issue.** When learning a new task, the parameter tensor will deviate from its optimal value for former tasks due to the increased knowledge related to current task. This process can be formally described as

$$
\begin{aligned}
W_t^* X_{t-1,i} &= \left(W_{t-1}^* + \sum_{z=1}^{Z} \Delta W_{t,z}\right) X_{t-1,i} \\
&= W_{t-1}^* X_{t-1,i} + \sum_{z=1}^{Z} \Delta W_{t,z} X_{t-1,i},
\end{aligned}
\tag{1}
$$

where $\sum_{z=1}^{Z} \Delta W_{t,z} X_{t-1,i}$ is the value of parameter shifting. $W_t^*$ is the optimal parameter under task $t$ and $Z$ is the total training epochs. $\Delta W_{t,z} = -\eta_{t,z} \nabla \mathcal{L}_{t,z}$ represents the parameter variation at epoch $z$ when learning task $t$. $\nabla \mathcal{L}_{t,z}$ is the gradient and $\eta_{t,z}$ is the learning rate.

**Existing Solution in FSCLP.** Eq.(1) implies that the condition $\sum_{z=1}^{Z} \eta_{t,z} \nabla \mathcal{L}_{t,z} X_{t-1,i} \rightarrow 0$ is required for a neural network to preserve the knowledge learned from previous tasks. Therefore, FSCLP methods [3, 20, 27, 28, 33, 37] project the gradients onto the orthogonal direction of task-specific feature space to ensure

$$
Proj_{X^\perp}(\nabla \mathcal{L}_{t,z}) X_{t-1,i} \rightarrow 0, \ z = 1, 2, \cdots, Z, \tag{2}
$$

where $X$ denotes the feature space and $X^\perp$ is its orthogonal direction. From Eq.(2), we observe that: (1) FSCLP aims to minimize the parameter shifting to maintain the performances on old tasks. This idea is closely related to stability, while it lacks consideration for plasticity. (2) Only previous features ($X_{t-1,i}$) are applied to construct the projection space for current gradients.

## 3.2 Gradient Projection in Common Null Space

GPCNS is different from existing FSCLP methods, because it is designed from the perspective of plasticity. Moreover, it utilize previous gradients rather than features to construct projection spaces. This idea stems from the observation of the parameter updating formula under gradient projection:

$$
W_t^* = W_1^* + \sum_{t=2}^{T} \eta_t Proj(\nabla \mathcal{L}_t), \ T \ge 2. \tag{3}
$$

We find that if the projected gradients under different tasks always appear in mutually orthogonal subspaces, then current gradients can be updated with confidence rather than considering mutual interference.

This idea can also be equivalently expressed as establishing an orthogonal set composed of the projected gradients under different tasks:

$$
\{\nabla \mathcal{L}_1, Proj(\nabla \mathcal{L}_2), \cdots, Proj(\nabla \mathcal{L}_{t+1})\}. \tag{4}
$$

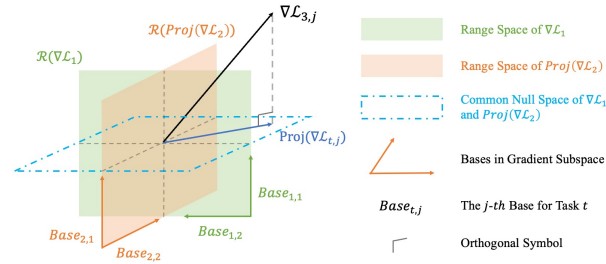

**Figure 2: Illustration of the common null space for the third task. For simplicity, we assume that there are only two bases in each subspace. The range space of the projected gradients in task 2 (red area) is orthogonal to the gradients in task 1 (green area), i.e. $\mathcal{R}(Proj(\nabla \mathcal{L}_2)) \perp \mathcal{R}(\nabla \mathcal{L}_1)$. The gradient generated in each epoch under task 3 will be projected into the common null space (see dashed area) for all previous tasks, namely tasks 1 and 2. This proposition can also be equivalently stated as the range space of $Proj(\nabla \mathcal{L}_3)$ is the common null space of $\nabla \mathcal{L}_1$ and $Proj(\nabla \mathcal{L}_2)$ according to Lemma 1 (see supplementary materials), i.e. $\mathcal{R}(Proj(\nabla \mathcal{L}_3)) = \mathcal{N}(\nabla \mathcal{L}_1) \cap \mathcal{N}(Proj(\nabla \mathcal{L}_2))$.**

Here $Proj(\nabla \mathcal{L}_t) \in \mathbb{R}^{m \times n}$ denotes the projected gradient under task $t$, and Eq.(4) can be further simplified as

$$
G_t \cdot Proj(\nabla \mathcal{L}_{t+1})^\top = 0, \tag{5}
$$

where $G_t \in \mathbb{R}^{tm \times n}$ is the common gradient matrix, formed by the vertical concatenation of the projected gradients from all previous tasks, namely:

$$
G_t = \begin{bmatrix} \nabla \mathcal{L}_1^* \\ Proj(\nabla \mathcal{L}_2^*) \\ \vdots \\ Proj(\nabla \mathcal{L}_t^*) \end{bmatrix} = \begin{bmatrix} G_{t-1} \\ Proj(\nabla \mathcal{L}_t^*) \end{bmatrix}. \tag{6}
$$

Here $\nabla \mathcal{L}_t^* \in \mathbb{R}^{m \times n}$ is the gradient under task $t$ at the end of training. We only take the final gradient because the projected gradients under the same task are in the same subspace, namely $\mathcal{N}(Proj(\nabla \mathcal{L}_t)) = \mathcal{N}(Proj(\nabla \mathcal{L}_t^*))$. We denote $\mathcal{N}(\cdot)$ and $\mathcal{R}(\cdot)$ as the null space and range space respectively, their definitions are presented in supplementary materials.

If Eq.(5) holds, the gradients under GPCNS will be projected into the **common null space** of $Proj(\nabla \mathcal{L}_t)$ under all previous tasks, namely $Proj(\nabla \mathcal{L}_{t+1}) \in \mathcal{N}(\nabla \mathcal{L}_1) \cap \mathcal{N}(Proj(\nabla \mathcal{L}_2)) \cap \cdots \cap \mathcal{N}(Proj(\nabla \mathcal{L}_t))$. To illustrate this idea, we demonstrate the projection space for the third task as an example in Figure 2, which is also the common null space for $\nabla \mathcal{L}_1$ and $Proj(\nabla \mathcal{L}_2)$.

To obtain the common null space, we perform SVD on $G_t$:

$$
G_t = [\widetilde{U}_t, \overline{U}_t] \begin{bmatrix} \Sigma_t & 0 \\ 0 & 0 \end{bmatrix} [\widetilde{V}_t, \overline{V}_t]^\top. \tag{7}
$$

Here $U_t = [\widetilde{U}_t, \overline{U}_t] \in \mathbb{R}^{tm \times tm}$, $V_t = [\widetilde{V}_t, \overline{V}_t] \in \mathbb{R}^{n \times n}$ and $\Sigma_t \in \mathbb{R}^{\min\{tm,n\} \times \min\{tm,n\}}$. Then we have $G_t \cdot \overline{V}_t = \widetilde{U}_t \Sigma_t (\widetilde{V}_t)^\top \overline{V}_t = \widetilde{U}_t \Sigma_t \cdot 0 = 0$, since $V_t$ is an orthogonal matrix. Thus, $\overline{V}_t \in \mathbb{R}^{n \times (n-tm)}$ is the null space of $G_t$ (see Definition 3 in supplementary materials). However, we do not adopt $\overline{V}_t$ because it will not exist if $tm \ge n$.

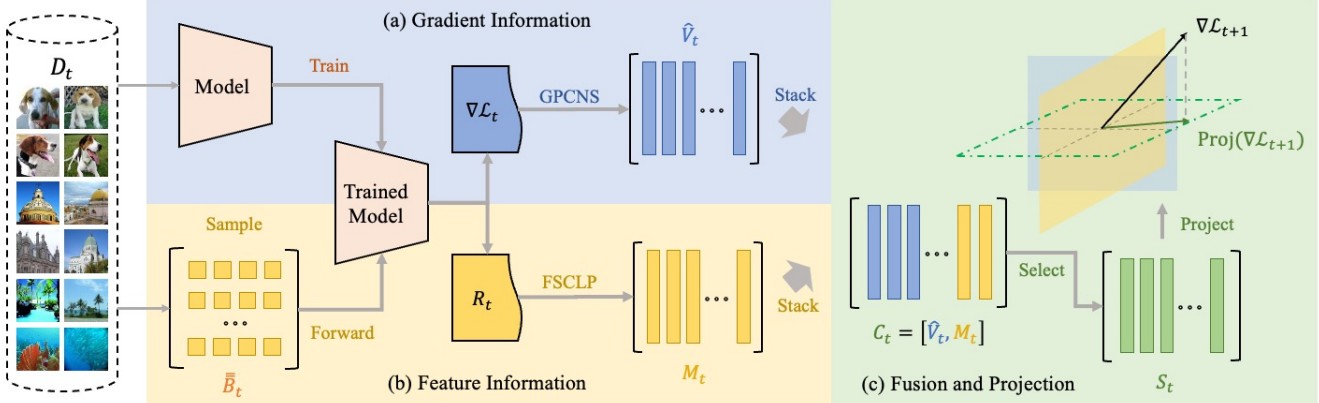

**Figure 3: The collaborative framework for FSCLP and GPCNS integrates gradient information and feature information from old tasks to determine the projection space. (a) Gradient Information: When learning task $t$, the training data $D_t$ is inputed into the model for training, and the gradient bases $\widehat{V}_t$ are obtained through step 10-14 in algorithm 1. (b) Feature Information: A sufficient number of samples are sampled from datasets for obtaining representation matrix, and feature bases $M_t$ are obtained through FSCLP methods. (c) Fusion and Projection: The bases of gradients and features are combined and filtered to form a matrix $S_t$ containing better bases. Finally, the gradients under next task will be projected into the null space of $S_t$.**

---

**Algorithm 1** Algorithm for GPCNS

**Input:** Datasets $\{\mathcal{D}_1, \mathcal{D}_2, \cdots, \mathcal{D}_T\}$ for each tasks; A neural network $f_W$ parameterized by $W$ with $L$ layers, learning rate $\eta$, threshold $\epsilon$ and scale coefficient $\alpha$.

**Output:** $f_W, S_T, \Lambda_T$;

1: initialization $f_W$
2: **for** $t = 1, 2, \cdots, T$ **do**
3:    **while** not converged **do**
4:       $B_t \sim \mathcal{D}_t$
5:       $\nabla \mathcal{L}_t \leftarrow Optimizer(B_t, f_W)$;
6:       $\nabla \widetilde{\mathcal{L}}_t \leftarrow Project(\nabla \mathcal{L}_t, S_{t-1}, \Lambda_{t-1})$    ▷ Eq.(11)
7:       $W_t \leftarrow W_t - \eta \nabla \widetilde{\mathcal{L}}_t$
8:    **end while**
9:    **for** $l = 1, 2, \cdots, L$ **do**
10:       $G_t^l \leftarrow VerticalStack\left(G_{t-1}^l, \nabla \widetilde{\mathcal{L}}_t^l\right)$   ▷ Eq.(6)
11:       $\Sigma_t^l, (\widetilde{V}_t^l)^\top \leftarrow \text{SVD}\left(G_t^l\right)$      ▷ Eq.(7)
12:       $\widetilde{V}_t^l \leftarrow Transpose((\widetilde{V}_t^l)^\top)$
13:       $k_t^l \leftarrow Criteria(\Sigma_t^l, \epsilon_t)$         ▷ Eq.(8)
14:       $\widehat{V}_t^l \leftarrow \widetilde{V}_t^l[:, 0 : k_t^l]$
15:       $H_t^l \leftarrow HorizonStack\left(H_{t-1}^l, \widehat{V}_t^l\right)$   ▷ Eq.(9)
16:       $\widetilde{U}_{t,H_t}^l, \Sigma_{t,H_t}^l \leftarrow \text{SVD}(H_t^l)$
17:       $k_{t,H_t}^l \leftarrow Criteria(\Sigma_{t,H_t}^l, \epsilon_t)$
18:       $S_t^l \leftarrow \widetilde{U}_{t,H_t}^l\left[:, 0 : k_{t,H_t}^l\right]$
19:       $\Lambda_t^l \leftarrow Scaling\left(\Sigma_{t,H_t}^l, \alpha\right)$   ▷ Eq.(10)
20:    **end for**
21: **end for**

---

Instead, $I - \widetilde{V}_t \widetilde{V}_t^\top \in \mathbb{R}^{n \times n}$ is also the null space of $G_t$ (see proof in supplementary materials), where $I \in \mathbb{R}^{n \times n}$ is the identity matrix.

Before constructing the projection operator, we select $k_t$ vectors in $\widetilde{V}_t$ through the criteria:

$$\|(G_t)_{k_t}\|_* \geq \epsilon_t \cdot \|G_t\|_*. \tag{8}$$

Here $\| \cdot \|_*$ denotes the nuclear norm (sum of the singular values) and $\epsilon_t \in [0, 1]$ is the threshold. $(G_t)_{k_t} \in \mathbb{R}^{k_t \times k_t}$ contains $k_t$ vectors with top-$k_t$ largest singular values. The bases contained in $\widetilde{V}_t$ will be selected as $\widehat{V}_t = \widetilde{V}_t[:, 0 : k_t] \in \mathbb{R}^{n \times k_t}$.

To provide GPCNS with more bases to choose from, we define the matrix $H_1 = \widehat{V}_1 \in \mathbb{R}^{n \times k_1}$ in task 1. When $t \geq 2$, we update the matrix $H_t$ by stacking $H_{t-1} \in \mathbb{R}^{n \times h_{t-1}}$ and $\widehat{V}_t$ horizontally:

$$H_t = \left[H_{t-1}, \widehat{V}_t\right] \in \mathbb{R}^{n \times (h_{t-1} + k_t)}, t \geq 2. \tag{9}$$

Next, SVD is performed on $H_t = U_{t,H_t} \Sigma_{t,H_t} V_{t,H_t}^\top$, where $U_{t,H_t} = \left[\widetilde{U}_{t,H_t}, \overline{U}_{t,H_t}\right] \in \mathbb{R}^{n \times n}$. The norm-based criteria $\|(U_{t,H_t})_{k_{t,H_t}}\|_* \geq \epsilon_t \|U_{t,H_t}\|_*$ is applied to select top-$k_{t,H_t}$ bases. The bases contained in $\widetilde{U}_{t,H_t}$ will be reduced as $S = \widetilde{U}_{t,H_t}[:, 0 : k_{t,H_t}] \in \mathbb{R}^{n \times k_{t,H_t}}$. Here we adopt $\widetilde{U}_{t,H_t}$ because it is the range space of $H_t$ (see proof in supplementary materials), which is also the range space of the common null space.

We scale the gradients in GPCNS through the following formula [28]:

$$\lambda_{t,i} = \frac{(\alpha + 1)\sigma_{t,i}}{\alpha \sigma_{t,i} + max(\sigma_t)}. \tag{10}$$

Here $\sigma_{t,i}$ is the $i$-th singular value of $H_t$ under task $t$, namely $\Sigma_{t,H_t} = \text{diag}([\sigma_{t,1}, \cdots, \sigma_{t,\min\{m,n\}}])$. The value of $\lambda_{t,i}$ are range from 0 to 1 with the above construction. $\lambda_t = \left[\lambda_{t,1}, \lambda_{t,2}, \cdots, \lambda_{t,n}\right]^\top$ is the importance vector under task $t$. We perform scaled gradient projection on gradient $\nabla \mathcal{L}_{t+1}$ as follow:

$$Proj(\nabla \mathcal{L}_{t+1}) = \left(I - S_t \Lambda_t (S_t)^\top\right) \nabla \mathcal{L}_{t+1}. \tag{11}$$

Here $\Lambda_t \in \mathbb{R}^{n \times n}$ is a diagonal matrix containing $\lambda_t$ in its diagonal. The main steps of our GPCNS are summarized in Algorithm 1.

## 3.3 Collaborative Framework between FSCLP and GPCNS

To construct projection spaces through both gradient and feature information, we propose a collaborative framework between FSCLP and GPCNS, which has two additional stages as following.

**1) Obtaining Bases Related to Features.** After obtaining $\widehat{V}_t \in \mathbb{R}^{n \times k_t}$ through step 10-14 in Algorithm 1, we further utilize GPM [27], TRGP [20] or SGP [28] as an algorithm under FSCLP to compute a memory matrix $M_t \in \mathbb{R}^{n \times \beta_t}$ which stores the bases related to features. Here $\beta_t$ is a positive integer, its value depends on how many bases are selected to represent the feature space.

The first dimension of $M_t$ is equal to $n$ because the gradient projection formula $\nabla \mathcal{L}_t = \nabla \mathcal{L}_t - \nabla \mathcal{L}_t M_t M_t^\top$ in FSCLP methods [3, 20, 27, 28] require that the rows of $M_t$ equals the columns of $\nabla \mathcal{L}_t \in \mathbb{R}^{m \times n}$.

**2) Integrating Gradient and Feature Information.** To integrate two types of information, we stack $\widehat{V}_t$ and $M_t$ horizontally:

$$C_t = \begin{bmatrix} \widehat{V}_t, M_t \end{bmatrix} \in \mathbb{R}^{n \times (k_t + \beta_t)}. \tag{12}$$

To filter the redundant bases, we perform SVD on $C_t$ and obtain $U_{t,C_t}, \Sigma_{t,C_t}, V_{t,C_t} = \text{SVD}(C_t)$, where $U_{t,C_t} = \begin{bmatrix} \widetilde{U}_{t,C_t}, \overline{U}_{t,C_t} \end{bmatrix} \in \mathbb{R}^{n \times n}$ is the left singular matrix of $C_t$, and $\widetilde{U}_{t,C_t} \in \mathbb{R}^{n \times \min\{n, (k_t + \beta_t)\}}$ is the range space of $C_t$ (see proof in supplementary materials). We select the vectors with top-$r_t$ largest singular values in $\widetilde{V}_{t,C_t}$ through the criteria $\|(C_t)_{r_t}\|_* \geq \epsilon_t \cdot \|C_t\|_*$ with the given threshold $\epsilon_t$. We only retain the first $r_t$ columns in $\widetilde{U}_{t,C_t}$, and the bases contained in $\widetilde{U}_{t,C_t}$ will be further reduced as $S_{t,C_t} = \widetilde{U}_{t,C_t}[:, 0 : r_t]$. Then, we also utilize the scaling matrix to adjust the projected gradient (see Eq.(10)), and the final gradient projection formula is

$$Proj(\nabla \mathcal{L}_{t+1}) = \left( I - S_{t,C_t} \Lambda_{t,C_t} S_{t,C_t} \right) \nabla \mathcal{L}_{t+1}. \tag{13}$$

The pipeline of our collaborative framework is demonstrated in Figure 3 and the main steps are summarized in Algorithm 2 (see supplementary materials).

## 4 Experiments

## 4.1 Experimental Setting

**Datasets.** We evaluate our methods on Split CIFAR-100 [14], CIFAR-100 Superclass [39] and Split MiniImageNet [31]. **Split CIFAR-100** contains 60,000 RGB images over 100 classes splitted into 20 tasks with 10 distinct classes per task. Each class contains 500 training images and 100 testing images, and the size of each image is 32 $\times$ 32. **CIFAR-100 Superclass** is divided into 20 tasks where each task contains 5 semantically related classes from CIFAR-100. **Split MiniImageNet** is a 100-class subset of the original ImageNet [6], which is splitted into 20 tasks. Each class contains 500 training images and 100 testing images. These images are in RGB format with 84 $\times$ 84 as their sizes.

**Implementation Details.** For the convenience of comparison, we adopt the same backbone with GPM, TRGP, and SGP on each dataset. Specifically, we apply a 5-layer AlexNet [15] on Split CIFAR-100, a LeNet on CIFAR-100 Superclass and a reduced ResNet-18 [11] on Split MiniImageNet to conduct experiments. All the methods are trained and tested under task-incremental learning setup that each task has a separate classifier head [12]. All the experiments are

conducted under the optimization of Stochastic Gradient Descent (SGD), and the batch sizes are set as 64. Each task in Split CIFAR-100 and Split MiniImageNet are trained for 200 epochs, and in CIFAR-100 Superclass are trained for 50 epochs. Other details are presented in supplementary materials.

**Baselines.** To follow the consistent experimental standards of GPM, TRGP, CGP and SGP, we require that the baselines not have the following situations: (1) Increasing parameters during the training process (e.g. API [17]). (2) Replacing cross entropy loss with customized objective functions (e.g. Connector [18] and DFGP [37]). (3) In addition, methods without open source code cannot be considered as baselines. (e.g. SD [45]). The reasons are as follows: (i) Increasing the number of parameters and replacing the loss function are essentially network expansion methods and regularization methods respectively. (ii) Our motivation is to improve gradient projection, rather than stacking different types of techniques. (iii) It is difficult to accurately reproduce methods without publicly available code.

Referring to the baseline selection in SGP [28], we select the following methods: OWM [40], A-GEM [1], Experience Replay with Reservoir sampling (ER_Res) [2], Adam-NSCL [33], GPM [27], FS-DGPM [5], CGP [3], TRGP [20] and SGP [28]. In addition, Multitask refers to learning all the tasks simultaneously, which can be seen as the upper bound of CL.

**Evaluation Metrics.** We employ average accuracy (ACC) and backward transfer (BWT) [23] as evaluation metrics. ACC represents average test accuracy across all tasks, and BWT measures the average decline in the test accuracy for previous tasks after learning the current task. Their definitions are as follow:

$$ACC = \frac{1}{T} \sum_{t=1}^{T} A_{T,t}; \quad BWT = \frac{1}{T-1} \sum_{t=1}^{T-1} A_{T,t} - A_{t,t}.$$

Here $T$ is the total number of tasks, and $A_{T,t}$ is the accuracy on $t$-th task after learning the $T$-th task sequentially.

## 4.2 Main Results

As shown in Table 1, introducing GPCNS can effectively improve the performances of FSCLP methods on all the benchmarks. Specifically, TRGP + GPCNS improves ACC by 1.12%, 1.26% and 4.28% compared with TRGP on CIFAR-100, Superclass and MiniImageNet respectively. GPM + GPCNS achieves the accuracy gains of 1.35%, 0.46% and 0.85% compared with GPM on these three datasets. Although GPCNS does not significantly improve the ACC of SGP, the improvements of FSCLP + GPCNS on BWT are more significant. TRGP + GPCNS improves BWT by 0.84%, 1.16% and 0.53% on three datasets. SGP + GPCNS achieves the BWT gains of 1.1%, 0.66% and 0.31% respectively. In addition, we find that even when GPCNS has higher accuracy than GPM or TRGP, its BWT is still lower. The reason might be that gradient matrices have much smaller size than representation matrices (see Table 1 in supplementary materials), which leads to less bases to be selected for balancing plasticity and stability when constructing projection spaces. From the above results, we can confirm that utilizing gradient and feature information simultaneously can indeed construct a better projection space, which improves ACC and further alleviates catastrophic forgetting compared to pure FSCLP methods.

 Chengyi Yang, Mingda Dong, Xiaoyue Zhang, Jiayin Qi, Aimin Zhou

**Table 1: Comparison results on several datasets. We report ACC and BWT over five runs with random seeds. The asterisk ∗ indicates a positive BWT on average.**

| Method | Split CIFAR-100 | | CIFAR-100 Superclass | | Split MiniImageNet | |
|---|---|---|---|---|---|---|
| | ACC (%) | BWT (%) | ACC (%) | BWT (%) | ACC (%) | BWT (%) |
| Multitask | 79.58 ± 0.54 | − | 61.00 ± 0.20 | − | 69.46 ± 0.62 | − |
| OWM [40] | 50.94 ± 0.60 | -30 ± 1 | − | − | 47.48 ± 1.28 | -12 ± 3 |
| A-GEM [1] | 63.98 ± 1.22 | -15 ± 2 | 50.35 ± 0.80 | -9.5 ± 0.9 | 57.24 ± 0.72 | -12 ± 1 |
| ER_Res [2] | 71.73 ± 0.63 | -6 ± 1 | 53.30 ± 0.70 | -3.4 ± 0.8 | 58.94 ± 0.85 | -7 ± 1 |
| Adam-NSCL [33] | 73.77 ± 0.50 | -1.6 ± 0.51 | 56.32 ± 0.88 | -2.42 ± 0.93 | 59.07 ± 1.10 | -4.9 ± 1.32 |
| GPM [27] | 72.48 ± 0.40 | -0.9 ± 0 | 57.72 ± 0.70 | -1.2 ± 0.4 | 60.41 ± 0.61 | -0.7 ± 0.4 |
| FS-DGPM [32] | 74.33 ± 0.31 | -2.71 ± 0.17 | 58.81 ± 0.34 | -2.97 ± 0.35 | 61.03 ± 1.08 | -1.96 ± 0.78 |
| CGP [3] | 74.26 ± 0.38 | -1.48 ± 0.78 | 57.53 ± 0.52 | -1.63 ± 0.49 | 60.82 ± 0.55 | -0.33 ± 0.21 |
| TRGP [20] | 74.46 ± 0.32 | -0.9 ± 0.01 | 58.25 ± 0.21 | -1.71 ± 0.52 | 61.78 ± 0.60 | -0.5 ± 0.6 |
| SGP [28] | 76.05 ± 0.43 | -1.23 ± 0.75 | 59.05 ± 0.21 | -1.4 ± 0.51 | 62.83 ± 0.33 | -1.12 ± 0.98 |
| GPCNS | 74.40 ± 0.42 | -2.16 ± 0.92 | 58.50 ± 0.43 | -1.86 ± 0.83 | 63.78 ± 0.62 | -2.84 ± 1.15 |
| GPM + GPCNS | 73.84 ± 0.29 | -0.26 ± 0.09 | 58.19 ± 0.38 | -0.47 ± 0.34 | 61.26 ± 0.44 | -1.25 ± 0.36 |
| TRGP + GPCNS | 75.58 ± 0.36 | **-0.06 ± 0.33** | **59.51 ± 0.32** | **-0.55 ± 0.27** | **66.07 ± 0.47** | *∗**0.03 ± 0.29** |
| SGP + GPCNS | **76.25 ± 0.38** | -0.13 ± 0.05 | 59.14 ± 0.40 | -0.74 ± 0.36 | 63.98 ± 0.53 | -0.81 ± 0.31 |

**Table 2: Ablation on gradient scaling (see Eq.(10)) in FSCLP + GPCNS and GPCNS. Here Feat and Grad are the abbreviation of Feature and Gradient respectively. The ticks below Feat or Grad indicate whether feature or gradient information is applied when constructing the projection spaces.**

| Method | Feat | Grad | CIFAR-100 | | Superclass | | MiniImageNet | |
|---|---|---|---|---|---|---|---|---|
| | | | ACC (%) | BWT (%) | ACC (%) | BWT (%) | ACC (%) | BWT (%) |
| SGP + GPCNS | ✓ | ✓ | **76.25 ± 0.38** | -0.13 ± 0.05 | 59.14 ± 0.40 | -0.74 ± 0.36 | 63.98 ± 0.53 | -0.81 ± 0.31 |
| SGP + GPCNS (w/o scaling) | ✓ | ✓ | 73.84 ± 0.29 | -0.26 ± 0.09 | 57.19 ± 0.38 | -0.47 ± 0.34 | 61.26 ± 0.44 | -1.25 ± 0.36 |
| TRGP + GPCNS | ✓ | ✓ | 75.58 ± 0.36 | **-0.06 ± 0.33** | **59.51 ± 0.32** | **-0.55 ± 0.27** | **66.07 ± 0.47** | *∗**0.03 ± 0.29** |
| TRGP + GPCNS (w/o scaling) | ✓ | ✓ | 74.21 ± 0.38 | -0.19 ± 0.36 | 59.17 ± 0.24 | -0.18 ± 0.21 | 63.28 ± 1.14 | -0.29 ± 0.14 |
| GPCNS | | ✓ | 74.40 ± 0.42 | -2.16 ± 0.92 | 58.50 ± 0.43 | -1.86 ± 0.83 | 63.78 ± 0.62 | -2.84 ± 1.15 |
| GPCNS (w/o scaling) | | ✓ | 73.46 ± 0.38 | -3.08 ± 1.12 | 58.39 ± 0.42 | -1.79 ± 0.72 | 61.90 ± 1.32 | -3.60 ± 1.34 |
| GPM [27] | ✓ | | 72.48 ± 0.40 | -0.9 ± 0 | 57.72 ± 0.70 | -1.2 ± 0.4 | 60.41 ± 0.61 | -0.7 ± 0.4 |

**Table 3: Ablation on $G_t$ and $H_t$ in GPCNS. Here Ulti Decom refers to the ultimately decomposed matrix to obtain the basis for constructing the projection space.**

| Method | Ulti Decom | CIFAR-100 | | Superclass | | MiniImageNet | |
|---|---|---|---|---|---|---|---|
| | | ACC (%) | BWT (%) | ACC (%) | BWT (%) | ACC (%) | BWT (%) |
| GPCNS | $H_t$ | 74.40 ± 0.42 | -2.16 ± 0.92 | 58.50 ± 0.43 | -1.86 ± 0.83 | 63.78 ± 0.62 | -2.84 ± 1.15 |
| GPCNS (w/o $H_t$) | $G_t$ | 73.91 ± 0.45 | -3.07 ± 1.11 | 58.18 ± 0.47 | -2.09 ± 0.96 | 61.42 ± 1.41 | -3.83 ± 1.29 |

## 4.3 Ablation Studies

**Ablation on Gradient Information and Feature Information.** To further verify the effectiveness of feature and gradient information, we conduct ablation experiments on task level, and the results are shown in Figure 4. On the top row, we compare FSCLP + GPCNS that performed better on different datasets with FSCLP methods, respectively. The results show that FSCLP + GPCNS has a visible improvement almost on each task for all the datasets because of additionally considering gradient information. The middle row displays the test accuracy curves of FSCLP + GPCNS and GPCNS for each task. Their fluctuations are similar, while FSCLP + GPCNS always has higher accuracy because of having found better bases for projection. When it comes to the bottom row, FSCLP + GPCNS has the highest BWT on all the datasets, indicating that the impact

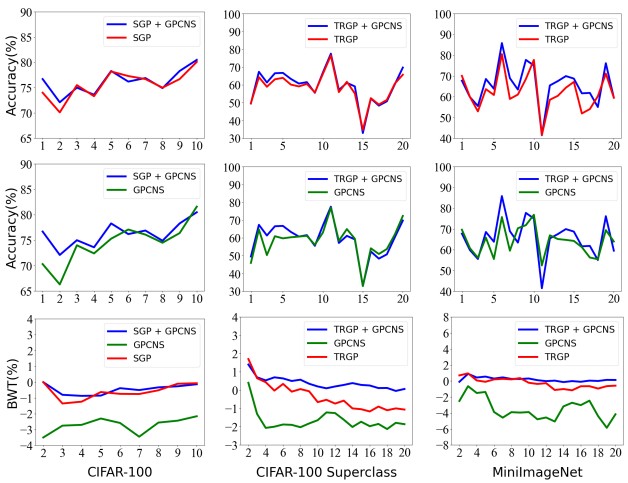

**Figure 4: Test accuracy and BWT on three datasets for each task. Top Row: Comparing accuracy between FSCLP + GPCNS and FSCLP. Middle Row: Comparing accuracy between FSCLP + GPCNS and GPCNS. Bottom Row: Comparison of BWT between GPCNS, FSCLP and FSCLP + GPCNS.**

of forgetting on this framework is minimal and its performances are stable.

**Ablation on Gradient Scaling.** We also conduct ablation experiments on gradient scaling. In Table 2, we observe that non-scaling methods reduce ACC by less than 0.5% on Superclass, and by less than 3% on CIFAR-100 as well as MiniImageNet compared to methods with scaling. In addition, we find that even if GPCNS does not utilize gradient scaling (see Eq.(10)), it still has higher accuracy than GPM. We know that both TRGP and SGP introduce gradient scaling on the basis of GPM. Therefore, gradient information and feature information are equally powerful and important.

**Ablation on $G_t$ and $H_t$ in GPCNS.** To verify the necessity of step 15-18 in Algorithm 1, we also conduct ablation experiments on $G_t$ and $H_t$ in Table 3. The results show that if $H_t$ is not utilized, the ACC will decrease by 0.49%, 0.32%, and 2.73% on CIFAR-100, Superclass and MiniImageNet respectively. Note that FSCLP + GPCNS adopts $G_t$ rather than $H_t$, since $M_t$ provides many additional bases for selecting.

**Ablation on Common Null Space.** To verify the effectiveness of projecting gradients into common null space, we replace the features in GPM with gradients for ablation experiments in Table 4. We observe that the accuracy of both GPM (Grad) and SGP (Grad) on CIFAR-100 Superclass collapse (see pale pink cells in Table 4). Therefore, FSCLP is not applicable to gradient information and proposing GPCNS with new paradigm is necessary.

**Ablation on norm type and top-$K$ signular value selection.** To show the effectiveness of adopting nuclear norm, we conduct ablation experiments by replacing nuclear norm with Frobius norm. To verify the necessity of selecting bases with top-$K$ signular value, we also ablate this step and conduct experiment on three dataset. The results in Table 5 show that either replacing the nuclear norm or excluding the top-$K$ singular values selection for filtering bases will result in a slight decrease in accuracy.

**Table 4: Ablation on Common Null Space. Here Grad with parentheses denotes that the features in GPM or SGP are replaced by gradients. In TRGP, features cannot be replaced with gradients because its code requires the size of feature matrices to remain unchanged.**

| Method | CIFAR-100 | | Superclass | | MiniImageNet | |
|---|---|---|---|---|---|---|
| | ACC(%) | BWT(%) | ACC(%) | BWT(%) | ACC(%) | BWT(%) |
| GPM | 72.48 | -0.9 | 57.72 | -1.2 | 60.41 | -0.7 |
| GPM (Grad) | 71.42 | -3.80 | 41.34 | -19.53 | 61.80 | -0.76 |
| SGP | 76.05 | -1.23 | 59.05 | -1.4 | 62.83 | -1.12 |
| SGP (Grad) | 68.72 | -8.18 | 40.83 | -20.12 | 61.64 | -1.28 |

**Table 5: Ablation on norm type and top-$K$ singular value selection. Here N-norm and F-norm are the abbreviation of nuclear norm and Frobius norm respectively. The values under ACC and BWT are both average percentages.**

| Method | CIFAR-100 | | Superclass | | MiniImageNet | |
|---|---|---|---|---|---|---|
| | ACC | BWT | ACC | BWT | ACC | BWT |
| GPCNS (N-norm) | 74.40 | -2.16 | 58.50 | -1.86 | 63.78 | -2.84 |
| GPCNS (F-norm) | 72.89 | -6.13 | 58.10 | -2.92 | 60.42 | -5.54 |
| GPCNS (w/o top-$K$) | 69.20 | -1.41 | 54.21 | -1.79 | 63.54 | -3.26 |
| TRGP + GPCNS (N-norm) | 75.58 | -0.06 | 59.51 | -0.55 | 66.07 | 0.03 |
| TRGP + GPCNS (F-norm) | 75.45 | -0.17 | 59.13 | -0.74 | 65.59 | -0.37 |
| TRGP + GPCNS (w/o top-$K$) | 72.75 | -0.09 | 56.78 | -0.06 | 64.96 | -0.25 |
| SGP + GPCNS (N-norm) | 76.25 | -0.13 | 59.14 | -0.74 | 63.98 | -0.81 |
| SGP + GPCNS (F-norm) | 75.12 | -1.68 | 58.87 | -1.42 | 61.54 | -2.33 |
| SGP + GPCNS (w/o top-$K$) | 74.09 | -0.32 | 57.96 | -1.88 | 62.41 | -1.06 |

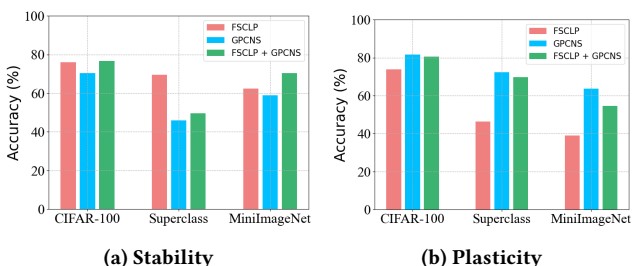

**(a) Stability**          **(b) Plasticity**

**Figure 5: The accuracy of final model. (a) Stability: Accuracy on the first task. (b) Plasticity: Accuracy on the last task. The FSCLP method on CIFAR-100 is SGP, and on Superclass and MiniImageNet is TRGP.**

## 4.4 Analysis on Stability and Plasticity

To verify the better plasticity of GPCNS and its effect on improving plasticity for FSCLP methods when used as a plugin, we record the test accuracy of the final model on the first and last tasks. In Figure 5b, we can observe that GPCNS always has the highest accuracy on the last task, which validates its advantage on plasticity. Correspondingly, we can also observe that FSCLP method has better stability than GPCNS in Figure 5a. More importantly, FSCLP + GPCNS leverages the strengths of both techniques in terms of stability and plasticity, resulting in improved ACC and BWT.

 Chengyi Yang, Mingda Dong, Xiaoyue Zhang, Jiayin Qi, Aimin Zhou

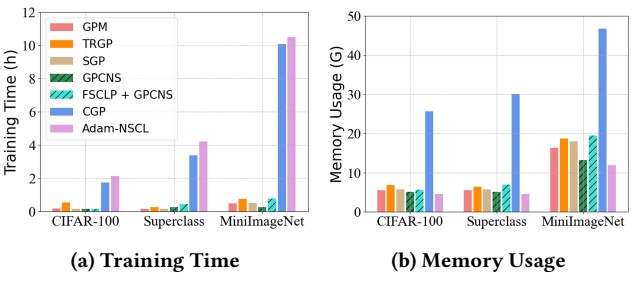

(a) Training Time                          (b) Memory Usage

**Figure 6: Total training time and memory usage at last epoch. The colors in (b) represent the same methods as in (a). Here FSCLP on CIFAR-100 is SGP, and on Superclass and MiniImageNet is TRGP.**

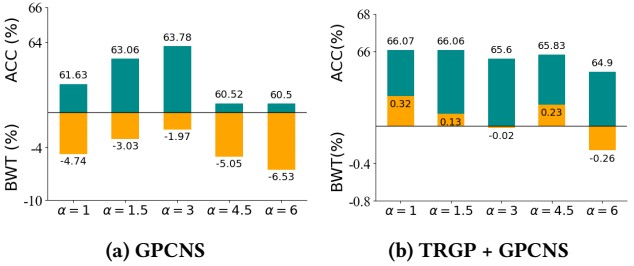

(a) GPCNS                                  (b) TRGP + GPCNS

**Figure 7: The impact of $\alpha$ when ResNet-18 with GPCNS and TRGP + GPCNS are performed on Split MiniImageNet.**

## 4.5 Training Time and Memory Overhead

In Figure 6, we compare the time consumption and memory usage of GPCNS and FSCLP + GPCNS with baseline methods. GPCNS spends the least training time on Split MiniImageNet and Split CIFAR-100, and also maintains a relatively low level on CIFAR-100 Superclass. In addition, we observe that the increased times of FSCLP + GPCNS due to simultaneously processing gradient and feature information are quite limited. In terms of memory, GPCNS has the second lowest memory overhead on all three datasets, and the memory usage of FSCLP + GPCNS has only slightly increased compared to pure FSCLP method.

The above results show that the computational costs of our GPCNS and FSCLP + GPCNS are maintained at a relatively low level. The reasons are as follows: (1) Only the final gradient of each task is required in both methods. (2) The sizes of gradient matrices stored and decomposed in our methods are generally much smaller than the sizes of representation matrices applied in GPM [27], CGP [3], TRGP [20] and SGP [28] (see Table 1 supplementary materials).

## 4.6 Hyperparameter Analysis

We vary the value of the scale coefficient $\alpha$ in Eq.(10). Specifically, $\alpha$ is set as $\{1.0, 1.5, 3.0, 4.5, 6.0\}$ successively. Figure 7 shows ACC and BWT when GPCNS and TRGP + GPCNS are performed on Split MiniImageNet. We observe that the optimal $\alpha$ for GPCNS and TRGP + GPCNS on Split MiniImageNet are 3 and 1, respectively. More significantly, TRGP + GPCNS achieves positive BWT when $\alpha$ equals 1, 1.5 and 4.5, which indicates that the old knowledge can

**Table 6: Comprehensive comparison between FSCLP and GPCNS. Note that the training cost in the table refers to when the number of tasks is not very large (e.g. 20 tasks).**

|  | FSCLP | GPCNS |
|---|---|---|
| **Information Source Design Idea** | Previous features. Minimizing the shifting of optimal parameters for previous task. | Previous gradients. Gradients under different tasks are projected in mutually orthogonal subspaces. |
| **Projection Space** | Orthogonal directions of previous feature subspace. | Common null space of the gradients for all previous tasks. |
| **Training Cost** | Slightly higher. | Slightly lower. |
| **Strengths** | Stability. | Plasticity. |
| **Weaknesses** | Plasticity. | Stability. |

be effectively transfered to new knowledge with the help of both gradient and feature information.

## 5 Discussion

**Comparison between FSCLP and GPCNS.** In Table 6, we compare our GPCNS with FSCLP methods, which demonstrates significant differences between GPCNS and previous gradient projection methods. GPCNS projects the gradients of previous tasks into the common null space, effectively utilizing gradient information and achieving better plasticity.

More importantly, the advantage of GPCNS happens to be what the FSCLP method lacks. Therefore, utilizing GPCNS as a plugin along with the previous FSCLP method can make a gradient projection method more comprehensive and powerful.

## 6 Conclusion

In this work, we analyze the effectiveness of FSCLP and find the limitations of FSCLP are neglecting gradient information and lacking plasticity. Then we propose the idea that gradients for different tasks should exist in mutually orthogonal subspaces, and further derive the concept of common null space. We propose GPCNS that projects current gradients into the common null space of the final gradients under all previous tasks. More importantly, we present a collaborative framework that GPCNS is served as a plug-and-play extension for FSCLP methods, enabling the simultaneous utilization of gradient and feature information. Finally, extensive experiments show that GPCNS exhibits higher plasticity that FSCLP methods lack. When GPCNS and FSCLP methods are employed under our collaborative framework, GPCNS can compensate for the shortcomings of existing methods in terms of plasticity and the utilization of gradient information, thereby constructing improved projection spaces and achieving enhanced performances.

## Acknowledgments

This work is supported by National Natural Science Foundation of China (No. 72293583, No. 72293580), Science and Technology Commission of Shanghai Municipality Grant (No. 22511105901) and Sino-German Research Network (GZ570).

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
