# OpenReview forum: "Introducing Common Null Space of Gradients for Gradient Projection Methods in Continual Learning"
_acmmm.org/ACMMM/2024/Conference — MM2024 Poster_

### Official Review · Reviewer_KWJo · 2024-05-26

**Rating:** 4
**Confidence:** 4

**Summary:**

This paper proposes Gradient Projection in Common Null Space to project current gradients into the common null space of final gradients under all preceding tasks to enhance plasticity and effectively utilize the gradients from old tasks. In addition, this method is plug-and-play, it can cooperate with other gradient projection methods to learn the valuable information contained in previous gradients. The proposed method outperforms all the gradient projection methods without increasing learnable parameters and customized objective functions. Ablation studies are also presented, showing the significance of each proposed technique.

**Strengths:**

1. The authors propose a novel gradient projection method to use the gradient of the current task to make the model learn more knowledge, enhancing plasticity.
2. This method can cooperate with existing gradient projections to learn more knowledge.
3. Extensive evaluations indicate that this method significantly outperforms previous gradient projection methods.

**Limitations:**

1. In this paper, the bases of gradient space and feature space are concatenated simply. Have you tried another effective way to fuse them?
2. Can these gradient projection methods be transferred into transformer architecture?
3. Can you provide the forward transfer metric to validate the plasticity of your method?

**Suitability:**

2

---

### Official Review · Reviewer_pz6H · 2024-05-27

**Rating:** 5
**Confidence:** 3

**Summary:**

The paper aims to enhance plasticity and effectively utilize the gradients from old tasks in Continual learning field,  and propose Gradient Projection in Common Null Space (GPCNS), which projects current gradients into the common null space of final gradients under all preceding tasks. Moreover, to integrate both feature and gradient information, we propose a collaborative framework that allows GPCNS to be utilized in conjunction with existing gradient projection methods as a plugin that provides gradient information and better plasticity.

**Strengths:**

The main contribution of this paper is to propose a new continuous learning strategy, gradient common null space projection (GPCNS), which aims to solve the problem that existing methods ignore gradient information and plasticity when dealing with catastrophic forgetting. By projecting the current gradient into the common null space of the final gradient of all previous tasks, the authors effectively utilize the gradient information of the old tasks and improve the plasticity of the model. In addition, the paper proposes a collaborative framework that combines GPCNS with an existing feature space-based Continuous Learning paradigm (FSCLP) to integrate feature and gradient information to build better projection Spaces. The experimental results show that GPCNS is superior to traditional methods in terms of plasticity, and can significantly improve the average accuracy (ACC) and backward transfer (BWT) of existing methods when applied as plug-ins.

**Limitations:**

1. Uncertainty of generalization ability
The paper mainly validates the effectiveness of GPCNS on specific benchmark datasets, but does not fully explore the generalization ability of the method for a wider range of tasks and more complex scenarios.
2. Hyperparameter adjustment:
Although the paper mentions hyperparameter α has a significant effect on model performance, but there is no systematic method or theoretical guidance to select the optimal hyperparameter values.
3. Cost calculation considerations:
The paper points out that GPCNS has advantages in training time and memory usage, but the computational cost of gradient matrix accumulation has not been fully evaluated as the number of tasks increases.

**Suitability:**

2

---

### Official Review · Reviewer_Rsms · 2024-05-30

**Rating:** 3
**Confidence:** 3

**Summary:**

The paper introduces a method designed to enhance continual learning by mitigating catastrophic forgetting and improving plasticity. It achieves this by projecting current gradients into the common null space of final gradients from all previous tasks. The proposed method, GPCNS, can be integrated as a plugin with existing gradient projection methods, providing a collaborative framework that enhances both the stability and plasticity of learning systems. The evaluation on three benchmarks demonstrates that GPCNS outperforms traditional methods by improving backward transfer and average accuracy without increasing the number of learnable parameters or requiring customized objective functions.

**Strengths:**

The concept of using common null space of gradients to project current task gradients is novel and addresses the plasticity issue in continual learning.

The paper provides rigorous mathematical proofs and a clear geometric interpretation of the common null space approach.

Extensive experiments across multiple benchmarks demonstrate the effectiveness of GPCNS in improving both backward transfer and average accuracy.

The authors have provided the code, which is beneficial for reproducibility and further research by the community.

The paper is well-written, with clear explanations of the methods and results. It also provides detailed diagrams and algorithms that aid in understanding the proposed methods.

**Limitations:**

While the paper demonstrates improvements over existing methods, it lacks a deeper analysis of how the approach compares against more recent or less conventional continual learning strategies.

The paper does not discuss the scalability of the method in detail, especially in scenarios with a very large number of tasks, which could be a limitation in practical applications.

The method's performance may depend heavily on the correct setting of hyperparameters like the scale coefficien.

Although the proposed method introduces novel ideas, the performance gains reported in the paper are relatively modest. The ablation studies, especially those pertaining to the gradient scaling and use of common null space, show only incremental improvements. This suggests that while the method is theoretically sound, its practical impact might be limited.

The paper lacks a comprehensive review of existing works that utilize the null space of gradients approach. This oversight is significant because it fails to position the proposed method within the broader context of the field. A thorough comparison with state-of-the-art methods, especially those that also leverage null space concepts in continual learning, is necessary to truly validate the novelty and effectiveness of GPCNS.
To address the gaps in literature review and comparative analysis, the authors should include a broader range of baseline methods, particularly focusing on those that employ similar strategies involving the null space of gradients, such as,
PNSP: Overcoming catastrophic forgetting using Primary Null Space Projection in continual learning. PRL 2024.
Balancing stability and plasticity through advanced null space in continual learning. ECCV 2022.

**Suitability:**

3

---

### Meta-Review · Area_Chair_pXvv · 2024-07-03

**Recommendation:** Accept (Poster)
**Confidence:** 4

**Metareview:**

The paper introduces a method designed to enhance continual learning by mitigating catastrophic forgetting and improving plasticity. Pre-rebuttal this paper received diverse ratings, i.e. 1 WA, 1 BA, and 1BR. The main concerns were lack of a deeper analysis (R1), insignificant results (R1), missing discussion of scalability  issue (R1), Uncertainty of generalization ability (R2). Post-rebuttal the paper has consistent ratings: 1 WA and 2 BA. The rebuttal addressed the concerns of the reviewers. AC agrees with the reviewers and tends to accept this paper.

---

### Meta-Review · Senior_Area_Chairs · 2024-07-10

**Recommendation:** Accept (Poster)
**Confidence:** 4

**Metareview:**

This paper received mixed ratings initially. After rebuttal, all reviewers tend to accept the paper. SAC and AC agree with reviewers and recommend accptence of the paper.